# Inter-Calibration of nine UV sensing instruments over Antarctica and Greenland since 1980

Clark Weaver [1,2], P. K. Bhartia[1], Dong L. Wu[3] , Gordon Labow[1,4] , David Haffner [1,4]

[1]Atmospheric Chemistry and Dynamics Branch, NASA Goddard Space Flight Center, Greenbelt, MD 20771, USA.

[2] Earth System Science Interdisciplinary Center (ESSIC), University of Maryland College Park, MD 20742, USA

[3]Climate and Radiation Laboratory, NASA Goddard Space Flight Center, Greenbelt, MD 20771, USA.

[4]Science Systems and Applications (SSAI), Inc., Lanham, MD 20706, USA.

*Correspondence to:* Clark Weaver clark.j.weaver@nasa.gov

**Abstract**
Nadir viewed intensities (radiances) from nine UV sensing satellite instruments are calibrated over the
East Antarctic Plateau and Greenland during summer. The calibrated radiances from these UV
instruments ultimately will provide a global long-term record of cloud trends and cloud response from
ENSO events since 1980. We first remove the strong solar zenith angle dependence from the intensities
using an empirical approach rather than a radiative transfer model. Then small multiplicative
adjustments are made to these solar zenith angle normalized intensities in order to minimize differences
when two or more instruments temporally overlap. While the calibrated intensities show a negligible
long-term trend over Antarctica, and a statistically insignificant UV albedo trend of -0.05 % per decade
over the interior of Greenland, there are small episodic reductions in intensities which are often seen by
multiple instruments. Three of these darkening events are explained by boreal forest. Other events are
caused by surface melting or volcanoes. We estimate a 2-sigma uncertainty of 0.35% for the calibrated
radiances.
**1. Motivation**
In 1980 the Nimbus-7 spacecraft carried the first **S**olar **B**ackscatter in the **UV** (SBUV) instrument into
low earth orbit to measure total column ozone. Since then, NOAA has deployed a suite of SBUV-2
instruments on board the NOAA-9, 11, 14, 16, 17, 18 and 19 spacecrafts. Since they were all nadir
viewing and thus had limited spatial coverage, NASA also deployed a suite of mapping instruments:
Nimbus-7 TOMS (1980), Earth Probe TOMS and the Nadir Mapper (NM) instrument of the Suomi
NPP Ozone Mapping Profiler Suite (OMPS, 2012). True to their design, they have provided a long-term
satellite data record of ozone products; however, they also were intended to measure the earth's
reflectivity in the UV at wavelengths insensitive to ozone (331 and 340nm). Aside from a few
publications (Herman et al. (2013), Labow et al. (2011) and Weaver et al. (2015)), this data set has not
been fully exploited. Our ultimate goal is a long-term record of a UV cloud product that can be directly
compared with climate models. This paper details the first step: the inter-calibration of radiances from
the suite of nadir viewing instruments. The second step retrieves a Black-sky cloud albedo (BCA)
record from the inter-calibrated intensities (Weaver et al. 2020) and compares the BCA with the
Shortwave CERES cloud albedo.
**2. Previous calibration of UV Satellite records**
The backbone of our data record is the suite of eight SBUV instruments starting with the Nimbus-7 in
1980, and ending with NOAA-19 in 2013.  Thereafter we use Nadir Mapper (NM) instrument on the
Suomi NPP OMPS. Each instrument provides narrowband backscattered intensities near the 340 nm
wavelength. We use a radiative transfer model to account for the small differences in each instrument's
center wavelength (see Appendix). Regular sun-viewing irradiance measurements ($F_{sun}$) are made,
typically weekly, to provide long-term calibration information. The measured intensities are normalized
by $F_{sun}$, and multiplied by $\pi$. Throughout this study **I** refers to the sun normalized intensities.
We start with intensities that have already been calibrated to account for instrument effects such as
hysteresis (see Deland et al 2012), and that are reported in the Level-2 datasets for each instrument. The
first seven SBUV/2 data sets were previously calibrated by characterizing the instruments over the East
Antarctic Plateau ice sheet using **L**ambertian **E**quivalent **R**eflectivity (LER, Huang L.-K. et al. 2003
and Herman et al. 2013). Using a radiative transfer model to calculate LER from the observed
intensities removes much of the solar zenith angle ($\boldsymbol{\theta_o}$) dependence, but not all; over the ice sheets LER
still decreases with $\boldsymbol{\theta_o}$ especially at high $\boldsymbol{\theta_o}$. While they did an excellent job of characterizing the first
seven SBUV/2 instruments, two additional sensors need to be intercalibrated to extend our record
forward: the SBUV2 on NOAA-19, and the Suomi NPP OMPS. Rather than calibrate these additional
instruments with a radiative transfer model using LER, we use an empirical approach to remove the
solar zenith angle dependence on intensity. Using these $\boldsymbol{\theta_o}$-normalized intensities, we inter-calibrate the
UV sensors over the East Antarctic Plateau and the Greenland ice sheets.
**3. Empirically based inter-calibration**
Satellite observed Nadir-viewed intensities over the Antarctic and Greenland ice sheets have an almost
linear relationship with solar zenith angle that is easily fitted with a 5-degree polynomial. Figure 1
shows the relationship over both ice sheets for all observations sampled by the SBUV2 on NOAA-16.
With a drifting orbit and long lifetime (2001-2014) NOAA-16 sampled a wide range of solar zenith
angles so we choose it as our reference instrument. The polynomial fit uses all observations over the
instrument's 14 year lifetime and so provides a most probable intensity that the NOAA-16 SBUV2
would observe for a given $\boldsymbol{\theta_o}$. Our calibration approach is to remove the solar zenith angle dependence
from the observed intensities ($\mathbf{I_{obs}}$) by using the reference polynomial fits shown in Figure 1. We can
test if an observed intensity is high or low compared with the NOAA-16 SBUV2 reference by
calculating a fractional deviation in terms of intensity ($\boldsymbol{\delta I}$) from Equation 1. For example, the right
panel of Figure 1 shows an anomalously low intensity sampled over a dark scene ($\mathbf{I_{obs}}$ dark scene)
observed at a solar zenith angle ($\boldsymbol{\theta}$o dark scene) ; it is compared with the intensity that NOAA-16 would
likely have observed at that solar zenith angle ($\boldsymbol{\xi}$ ($\boldsymbol{\theta}$o dark scene)) . The difference is divided by $\boldsymbol{\xi}$ ($\boldsymbol{\theta}$o dark
scene) to produce a fractional deviation in intensity $\boldsymbol{\delta I}$ which is common throughout the manuscript.
$$\boldsymbol{\delta I} = \frac{\mathbf{I_{obs}} - \xi(\theta_o)}{\xi(\theta_o)} \qquad\qquad \text{Equation 1}$$
Each UV instrument has its own unique $\mathbf{I_{obs}}$ to $\boldsymbol{\theta}$o relationship mainly because the photomultiplier tube
(PMT) for each instrument has a slightly different response function. The underlying scene UV albedo
(averaged over an instrument's lifetime) could be slightly different for each instrument, which would
also change the $\mathbf{I_{obs}}$ to $\boldsymbol{\theta}$o relationship, but we expect the Antarctic plateau albedo to be stable over time.
The SBUV PMTs are designed to have a zero-offset bias, i.e. zero current response when there are zero
photon counts. Although the polynomial fit is not constrained to have $\mathbf{I_{obs}}$=0 at a solar zenith angle of
90o, it appears so, consistent with this instrument design (Figure 1).
We also show estimates of Intensity calculated by the radiative transfer model VLIDORT (**V**ector
**LI**nearized **D**iscrete **O**rdinate **R**adiative **T**ransfer package, Spurr, 2006). Here we assume Lambertian
surface albedo of .95, and Rayleigh atmosphere with surface pressure of 663 hPa. The number of half-
space quadrature streams is 40; the number of Stokes vector parameters is 3. At first glance the
VLIDORT simulation appears to simulate the observations (red trace Figure 1) and we considered using
the $\mathbf{I}$ to $\theta_o$ relationship simulated by VLIDORT as a reference (instead of using NOAA-16).  But closer
examination shows that the slope of the VLIDORT is shallow compared with the observations. The
resulting $\boldsymbol{\delta I}$ would still be slightly dependent on $\theta_o$ which would complicate the analysis.
Another, more sophisticated approach to validate sun-normalized radiances over ice sheets is described
in Jaross et al. (2008). They account for snow surface BRDF and off-nadir viewing angles. Nadir
330nm reflectances simulated using their snow BRDF model are 1% less than those assuming a
Lambertian surface at $\theta_o$ =70o; disparities are near zero at $\theta_o$ =50o. Our nadir observed $\boldsymbol{\delta I}$ is not
sensitive to solar azimuth angle over Antarctica.
The suite of SBUV/2 instruments provides nadir observations with a 170x170km **F**ield **O**f **V**iew (FOV).
But the OMPS Mapper instrument has a smaller nominal 50x50km FOV, except at the two most nadir
viewing positions. Here the FOV widths are 20 and 30 km (Seftor el al 2017). For consistency, we only
used the Mapper viewing positions that were within a nadir-centered hypothetical 170x170km SBUV
FOV and aggregated their intensities (area weighted) prior to calculating $\boldsymbol{\delta I}$. For each instrument we
calculate the summertime annual mean and plot the timeseries for both ice sheets (Figure 2).
**4. Adjusting the intensities**
The pre-calibrated intensities SBUV2 instruments on board NOAA-17, -18 and -19 appear to be high
biased compared to our reference (Figure 2). As described below, a cost-optimization approach is used
to adjust the intensities and reduce these disparities.  Figure 2 only shows the summertime average $\boldsymbol{\delta I}$,
but when calibrating instruments, it is instructive to examine the $\boldsymbol{\delta I}$ dependence on $\theta_o$ for individual

years. The left panel of Figure 3a shows this for 2006 when the reference and three other instruments were operational. The positive bias for NOAA-17 and 18 is consistent at all $\theta_o$ bins and suggests that a simple adjustment of the intensities might reduce these biases. All instruments show a similar skewed $\delta I$ distribution, at each $\theta_o$ bin, toward low values of $\delta I$.

To adjust intensities for a specific instrument a multiplicative factor ($c_1$) is chosen so that the adjusted intensities are a linear function of the original intensities: $\mathbf{I_{adj}} = c_1*\mathbf{I_{original}} + c_0$. Adjusting the multiplicative factor ($c_1$) changes the gain, (intensity per observed photon counts) of the instrument. To inter-calibrate all instruments with respect to NOAA-16 we use a minimum-cost optimization algorithm to solve for a set of $c_1$ values that minimizes $\delta I$ disparities between temporally overlapping instruments. The $c_1$ for each instrument, except the reference, is allowed to vary; Table 1 shows the gain changes made to each instrument. Note that $c_1$ does not depend on time, so the interannual variability of a specific SBUV instrument remains intact after the calibration.

Only the highest quality observations are used for the inter-calibration. Observations are limited to $\theta_o$ less than $75_o$ because at higher $\theta_o$ ozone absorption and straylight effects become significant and contaminate results. Furthermore, SBUV observations that have a grating drive error and observations that are likely impacted by PMT hysteresis are not used to intercalibrate.

The grating drive selects the wavelength of a SBUV measurement. Sometimes, but not too often, the grating drive selects the wrong value and the intensities are measured at a wavelength different than the

SBUV instrument's nominal wavelength. Inclusion of observations with uncorrected grating errors will
confuse our results, since our analysis assumes that intensities to derive $\delta\mathbf{I}$ are all at the same
wavelength. Fortunately, the grating drive position is archived so we can apply a correction (see
Appendix); however, the observations with uncorrected grating errors are not used in the
intercalibration, but are used in the later trend analysis. Figure 4 shows the summertime average
empirically adjusted $\delta\mathbf{I}$ over both ice sheets after applying the gain changes in Table 1. Solid circles
exclude observations with grating drive errors and open circles include corrected observations. There is
clearly tighter match between overlapping instruments compared with Figure 2. But there still are
disparities between overlapping instruments between 1997 and 1999 when multiple instruments suffer
from grating errors. It is disconcerting that our correction does not bring them in closer alignment.
Both Nimbus-7 and to a lesser extent NOAA-9, suffered from PMT hysteresis. These earlier PMTs
were not able to quickly respond to the 4 orders of magnitude signal changes that occur when the
satellite first comes out of darkness on each orbit and the instrument sees its first light. For Nimbus-7
hysteresis errors are between 4 and 9% at first light over Antarctica and lessen as the PMT adjusts to the
bright scenes over the ice sheet. By the time the Nimbus-7 reaches Greenland the PMT is equilibrated
and there is no hysteresis error. (Maximum hysteresis errors of NOAA-9 are 2%.). The intensity
observations for these early instruments have been corrected for hysteresis (Deland et al., 2001). Still,
we initially were unable to match Nimbus-7 with the other instruments; there was good agreement over
Antarctica but over Greenland Nimbus-7 was about 1% higher than the others (Figure 2).
Our remedy was to first calibrate the SBUV instruments *only* over Greenland where Nimbus-7 is free of
hysteresis error.  As expected, all temporally overlapping instruments agreed over Greenland, but over
Antarctica Nimbus-7 was low by about 1% compared with NOAA-9 and NOAA-11. Then we started
removing Nimbus-7 observations; first those within 1 minute of first light, then 2 minutes. With every
minute of observations removed, the disparity over Antarctica lessened.  We achieved the good
agreement seen in Figure 4 by removing 9 minutes of Nimbus-7 observations after first light.
Figure 5 shows the $\theta_o$ dependence on the empirically adjusted $\delta I$ for selected years. All the SBUVs,
except for Nimbus-7 and NOAA-9, have an almost flat (<0.005) $\delta I$ dependence with $\theta_o$. A flat $\theta_o$
dependence indicates that the PMT response is similar to the NOAA-16. Over Greenland $\delta I$
dependence with $\theta_o$ is not quite as flat (Figure 3b). The suppression of  $\delta I$ at $\theta_o > 57_o$ and time after
first light < 9 minutes is seen for all years of Nimbus-7. Even though these suppressed observations ($\theta_o$
> 57$_o$) were previously corrected for hysteresis, artifacts remain and they are not used in any analysis.
Multiple instruments show coincident reduction of $\delta I$ over Antarctica in January 1992 (Figure 4) most
likely from aerosols transported to the Antarctic after the eruption of Mt Pinatubo 6 months earlier in
1991 (left panel Figure 3c). The April 1982 eruption of El Chichon likely contributed to the coincident
reduction in 1983; other anomalies occur in 2001, 2010 and 2013. Likewise, there are coincident
reductions in $\delta I$ over Greenland.
To estimate the uncertainty in the SBUV intensity from instrument calibration alone we first average
the δI over the coincident satellites for each year; this merged time series represents the geophysical
contribution. Absolute departures from this merged time series (Figure 6) are attributed to instrument
calibration uncertainty. Two times the standard deviation of the fractional departures of all the SBUVs
and OMPS (using both ice caps) is about 0.0035. We conclude that annual averages of **I** have a 2-sigma
uncertainty of 0.35%.
**5. Greenland Ice Sheet**
The albedo of the Greenland Ice Sheet is of interest because it contributes to changes in the surface
energy balance and surface melting. The variability of our UV δI record is consistent with the MODIS
albedo data set. A recent study presents time series of the surface reflectance over the Greenland Ice
Sheet from the Collection 5 (C5) and C6 MODIS data sets (Casey et al. 2017). While the older C5 set
shows strong darkening of the ice sheet since 2000 (not shown), C6 has negligible trends that are not
statistically significant. They report surface reflectance for the channel closest to our UV channel
(MODIS Band 3, 459nm) for dry snow conditions (locations with ice surface elevations > 2000 m) and
for wet snow conditions (elevations < 2000 m). For easier comparison we have transcribed the data
from their Figure 4 onto our Figure 4c. Many of the same episodic events in the MODIS C6 record that
limit measurements to wet snow conditions (solid blue trace Figure 4c) are also seen by the UV
instruments (Figure 4b and c): darkening in the NH summer of 2003, 2010 and 2012. The 2012
darkening was likely driven by anomalous surface melting over Greenland. Satellite estimates of melt-
day area from microwave brightness temperatures (Nghiem et al., 2012) and mass loss from the NASA
GRACE instrument both suggest strong surface melting in 2012.
Surface or airborne light-absorbing aerosols that originate from boreal forest fires can explain some of
the other reductions of UV $\delta I$ over Greenland. The 1995 darkening episode is likely caused by forest
fires in Canada. Using a trajectory model, Wotawa and Trainer (2000) estimate that CO emitted from
the large fires in western Canada reach Greenland on 1 July (their figure 2). Using a similar technique,
Stohl et al 2006 estimate that CO from Alaskan and Canadian fires in 2004 reached Summit Greenland
on about 16 July. Their figure 11 shows elevated levels of observed and trajectory-modeled CO from 16
July to 2 August. Finally, the global travels of smoke from the 2003 fires in South eastern Russia are
documented by Damoah et al. (2004) using a trajectory model and MODIS satellite images. They
estimate a 24 May arrival time over Greenland (their Figure 2). A time-series of daily values of UV $\delta I$
over Greenland show abrupt reductions by the SBUV instruments operating on those dates (Figure 7).
There are other dramatic darkening events, likely caused by either forest fire smoke or surface melting
(e.g. 2006 and 2008), that we could not find in the literature.
While the shorter C6 record shows no apparent trend, our UV record shows a weak, though statistically
insignificant reduction in UV $\delta I$ over Greenland: -0.05 (+-0.06) decade$^{-1}$ at locations with elevations >
2000 meters (Figure 4b). Impurities in the snow as detected by insitu analysis are consistent with our
observed trend. Polashenski (et al 2015) measured the concentrations of light absorbing impurities
(LAI) in 67 snow pits across North West Greenland ice sheet in 2013 and 2014 and compared them
with studies that analyzed snow from the past 6 decades. Increases in black carbon or dust
concentrations relative to recent decades were small and corresponded to snow albedo reductions of at
most 0.31, or ~0.05 per decade which is similar to our UV satellite estimate. The snow studies also
record episodic events that darken the snow 1-2%, similar to the 1995, 2003 and 2004 darkening we see
in the SBUV satellite record.
**6. Discussion / Summary**
The East Antarctic Plateau is the preferred ice sheet for performing radiance calibration. Its very low
temperatures and clear pristine conditions, except for the occasional volcanic eruption, all maintain a
stable surface albedo with time. In contrast, the interior Greenland ice sheet is darkened every few years
by air-borne particles from Boreal wild fires or from albedo changes caused by widespread surface
melting. Since we are not doing an absolute calibration, but a relative calibration (using NOAA-16 as a
reference instrument), Greenland's albedo variations (~2%) test how well the SBUV instruments
respond to changes in the albedo. Moreover, including it in our calibration analysis enables a
characterization of instrument hysteresis errors mainly with Nimbus-7 over Antarctica. Once removed,
it matters little whether both ice sheets or only Antarctica are used to determine the multiplicative gain
coefficients ($c_1$), the UV $\delta I$ trends over both ice sheets are almost the same.
Intensities at the 340 nm wavelength channel observed by eight nadir-viewing SBUV satellite
instruments and the OMPS scanning instrument are intercalibrated over the Antarctic and Greenland ice
sheets. The approach is to compare observed intensities that have been normalized by solar zenith
angle. After the inter-calibration, we estimate a 2-sigma uncertainty of 0.35% based on temporally
overlapping sensors. Multiple instruments respond in unison to known darkening events that sometimes
can be explained by volcanic aerosols, soot from boreal forest fires, or surface meltwater. These
calibrated intensities will be used to derive a UV cloud albedo record over the tropics and midlatitudes
since 1980.
**Appendix - Accounting for small wavelength differences**
Each instrument provides narrowband backscattered intensities close to but not exactly at 340 nm
wavelength. For example, the Nimbus-7, NOAA-9 and NOAA-14 have nominal center wavelengths of
339.90, 339.75, 340.05 nm and Full Width Half Maximum (FWHM) of 1.0, 1.132 and 1.132nm,
respectively. These seemingly small wavelength differences will change observed intensities by several
tenths of a percent at high solar zenith angles. Using the VLIDORT Radiative Transfer Model we create
a 2-dimensional table of intensities at 0.1 nm wavelength resolution and at $10_o$ SZA resolution. A
Lambertian surface of 0.95 albedo is assumed. For each instrument we determine a simulated intensity
$I_{sim}$ by convolving the instrument's FWHM across the center wavelength of the instrument. To account
for the wavelength and FWHM difference between a non-reference instrument (e.g. Nimbus-7) and our
reference instrument NOAA-16 we multiply the observed intensities from Nimbus-7 by  $I(\theta_o)_{sim}$
$_{NOAA-16}$   $/ I(\theta_o)_{sim\ Nimbus-7}.$ Note that the wavelength correction is dependent on solar zenith angle.

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

## Linear equation for empirical $\delta I$ Adjustment

$$I_{adj} = c_1 * I_{original} + c_0$$

|  | $c_0$ | $c_1$ |
|---|---|---|
| Nimbus-7 SBUV | - | 0.9913 |
| NOAA-9 SBUV/2 | - | 1.0013 |
| NOAA-11 SBUV/2 | - | 1.0002 |
| NOAA-14 SBUV/2 | - | 1.0011 |
| NOAA-16 SBUV/2 | - | 1 |
| NOAA-17 SBUV/2 | - | 0.9962 |
| NOAA-18 SBUV/2 | - | 0.9936 |
| NOAA-19 SBUV/2 | - | 0.9976 |
| OMPS-Mapper | - | 0.9972 |

3    Table 1. Gain $c_1$ and offset $c_0$ values used to make adjustments to observed intensities for UV sensing
4    instruments.

## Figure 1

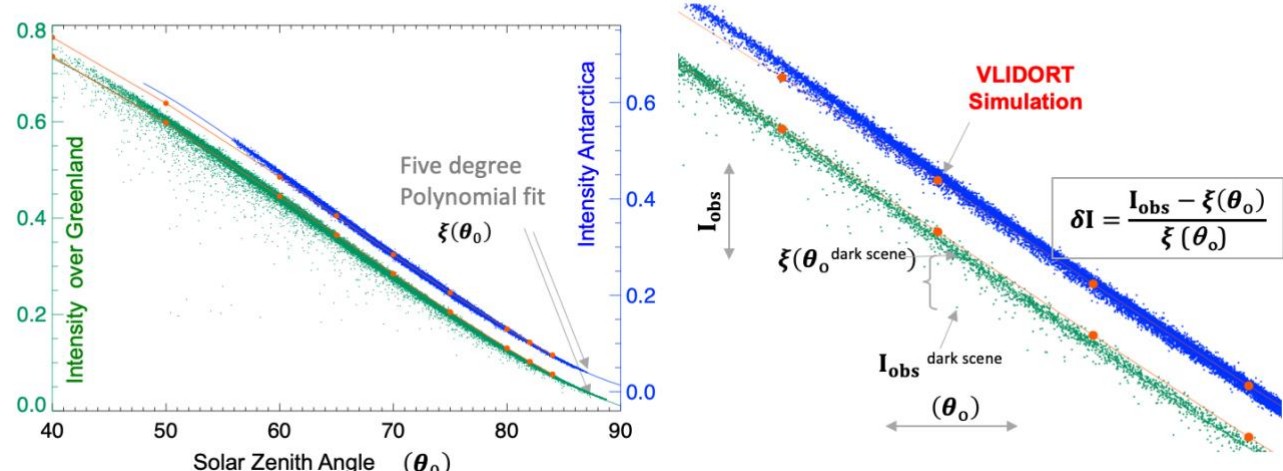

Figure 1. Measured Intensity at 340 nm from the NOAA-16 SBUV versus Solar Zenith Angle over the
Antarctic Plateau (blue) and Greenland (green). Each point is a nadir-viewed observation at the native
Field of View (170 km by 170 km) during the summer (fifteen days on either side of solstice). Also
shown is a polynomial fit and a radiative transfer simulation (red) assuming a Lambertian surface
albedo of .95, a Rayleigh atmosphere with surface pressure of 663 hPa. Note that the Greenland
intensities are offset from the Antarctic ones. The right panel shows a zoomed in view (see text for
details).

## Figure 2

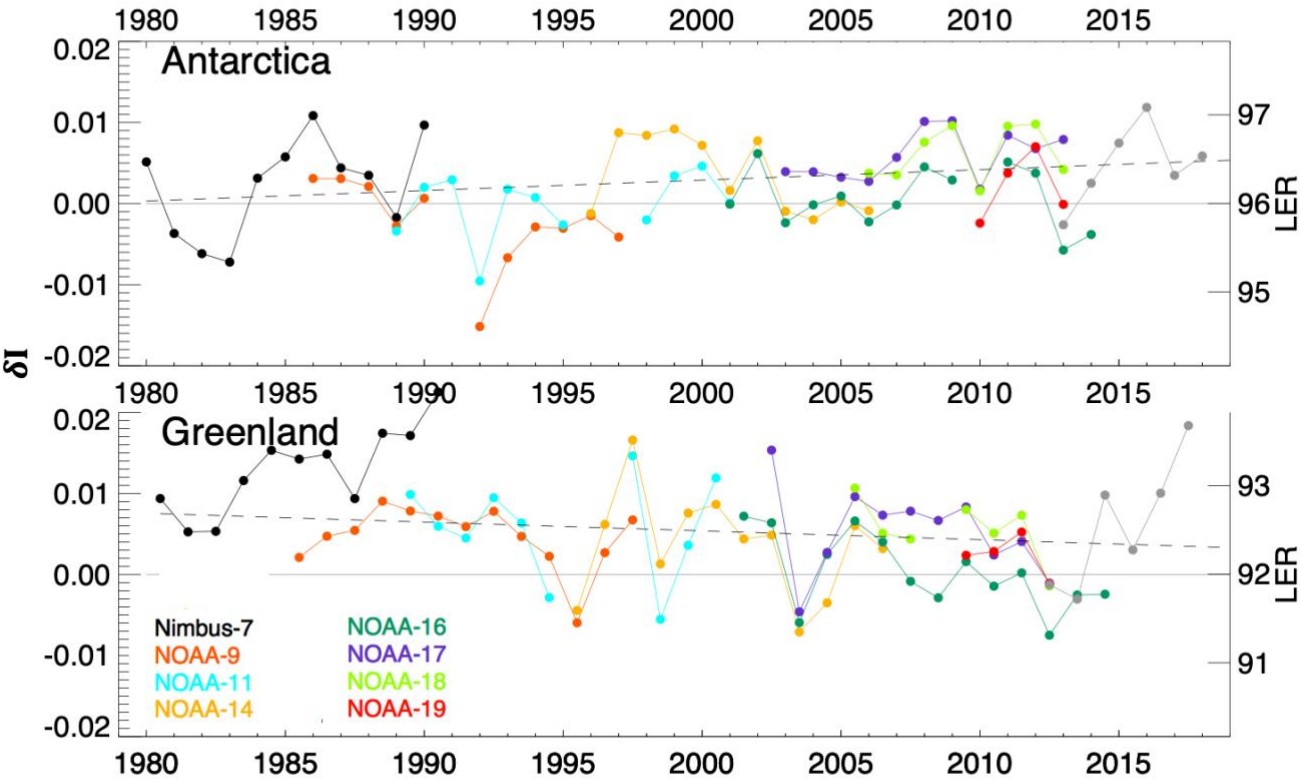

Figure 2. Inter-annual variability of previously calibrated **δI** for the SBUV instruments (colored) and OMPS mapper (grey) over Antarctica and Greenland.  The right-hand axis shows the corresponding change in LER.

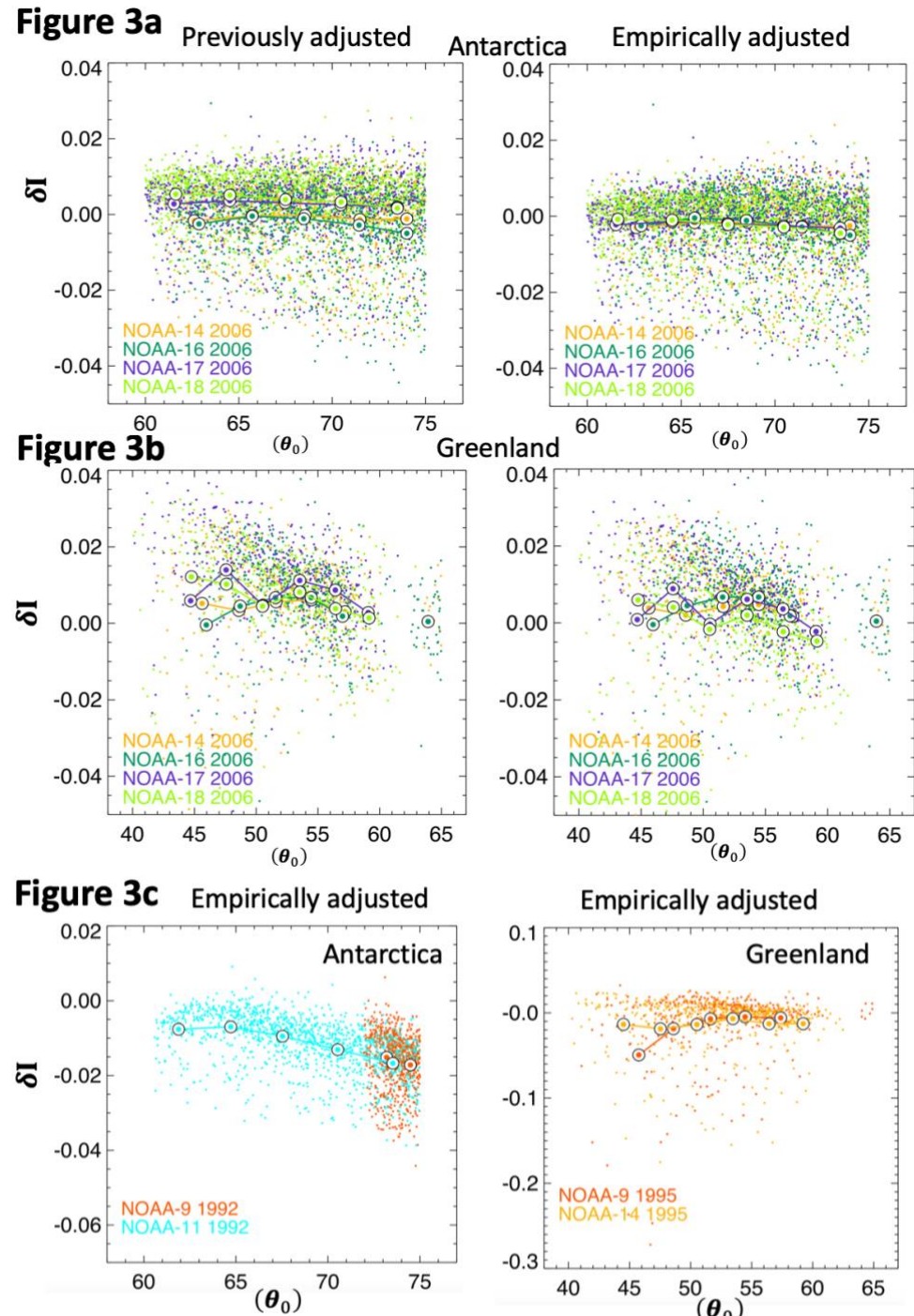

**Figure 3a** Previously adjusted    Antarctica    Empirically adjusted

NOAA-14 2006
NOAA-16 2006
NOAA-17 2006
NOAA-18 2006

**Figure 3b**    Greenland

NOAA-14 2006
NOAA-16 2006
NOAA-17 2006
NOAA-18 2006

**Figure 3c** Empirically adjusted      Empirically adjusted

Antarctica

Greenland

NOAA-9 1992
NOAA-11 1992

NOAA-9 1995
NOAA-14 1995

Figure 3. $\delta I$ for all FOVs observed over the ice sheets plotted against solar zenith angle ($\theta_o$) for specific years. The large circles are averages of $\delta I$ binned by solar zenith angle. Figure 3a shows the previously

calibrated δ**I** on the left and our empirically calibrated δ**I** over Antarctica on the right for 2006. Figure
3b is same but over Greenland for 2006. Figure 3c shows our empirically calibrated values over
Antarctica in 1992 and Greenland in 1995.

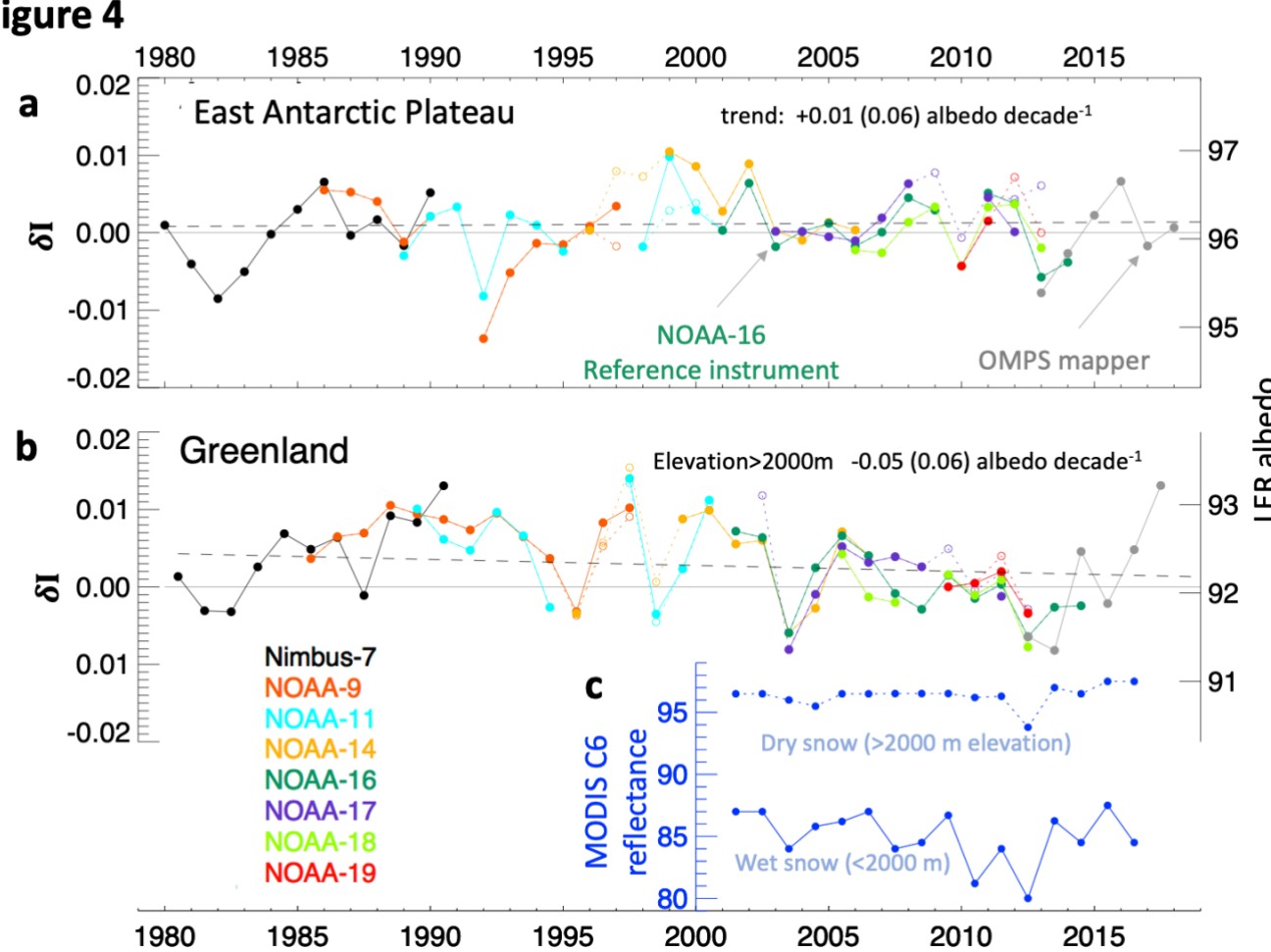

Figure 4. Inter-annual variability of our δ**I** for the SBUV instruments (colored) and the OMPS Mapper
(grey) over Antarctica (a) and interior locations over Greenland with ice surface elevations above 2000
meters (b). The right-hand axis shows the corresponding change in LER. Annual means plotted with
solid circles only include observations with correct grating drive positions; open circles also include
those with grating drive errors that have been corrected (see text). The lowest panel (Figure 4c) shows
MODIS Collection 6 reflectance for Band 3 (459nm) at elevations above 2000 meters (dry snow
conditions dashed trace) and below 2000 meters (wet snow conditions solid trace).

## Figure 5

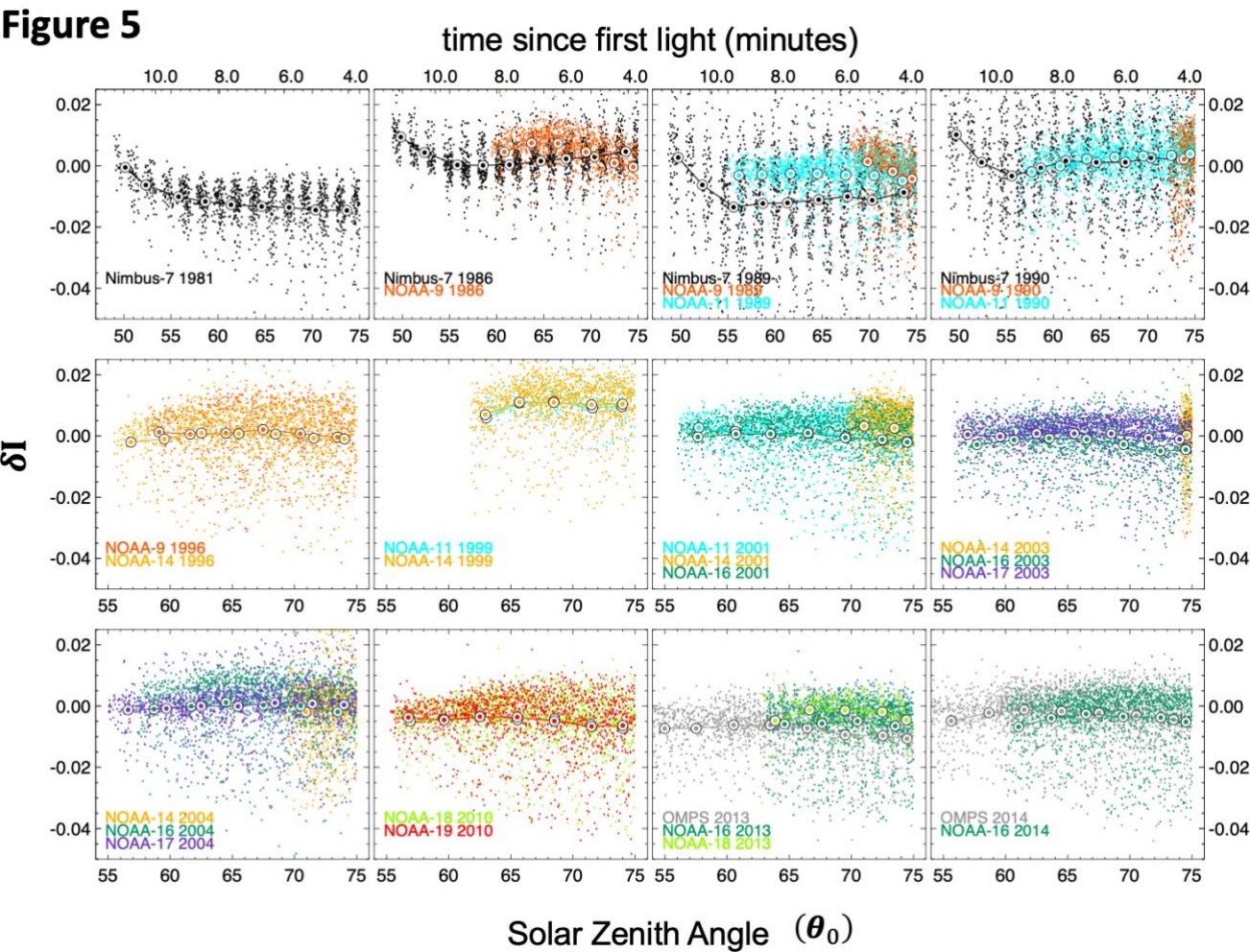

Figure 5. Empirically calibrated for all FOVs observed over Antarctica plotted against solar zenith angle ($\theta_o$) for selected years. The top four panels show the suppression of $\delta I$ during the first 7-10 minutes after Nimbus-7 sees its first light at the start of a new orbit. At first light, time=0 and $\theta_o$ =90o. The time after first light (minutes) is shown at top of first four panels.

**Figure 6**

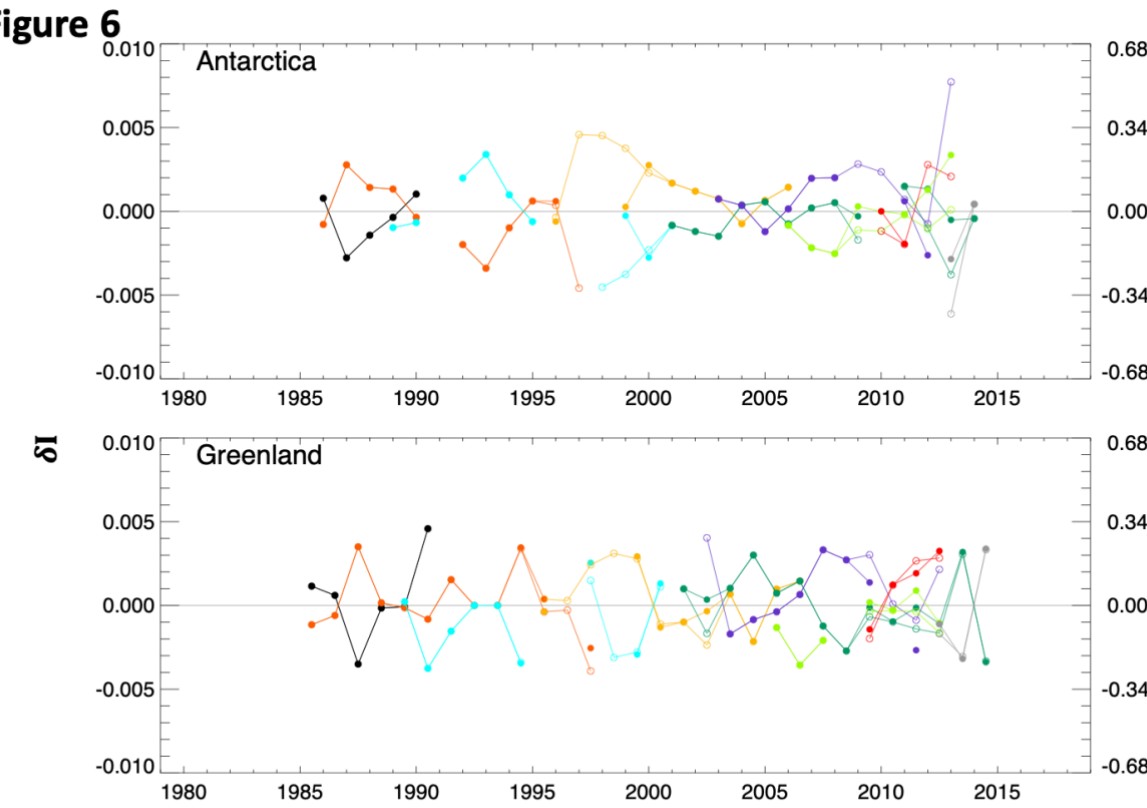

Figure 6. same as Figure 4 except that merged-satellite average is removed.

## Figure 7

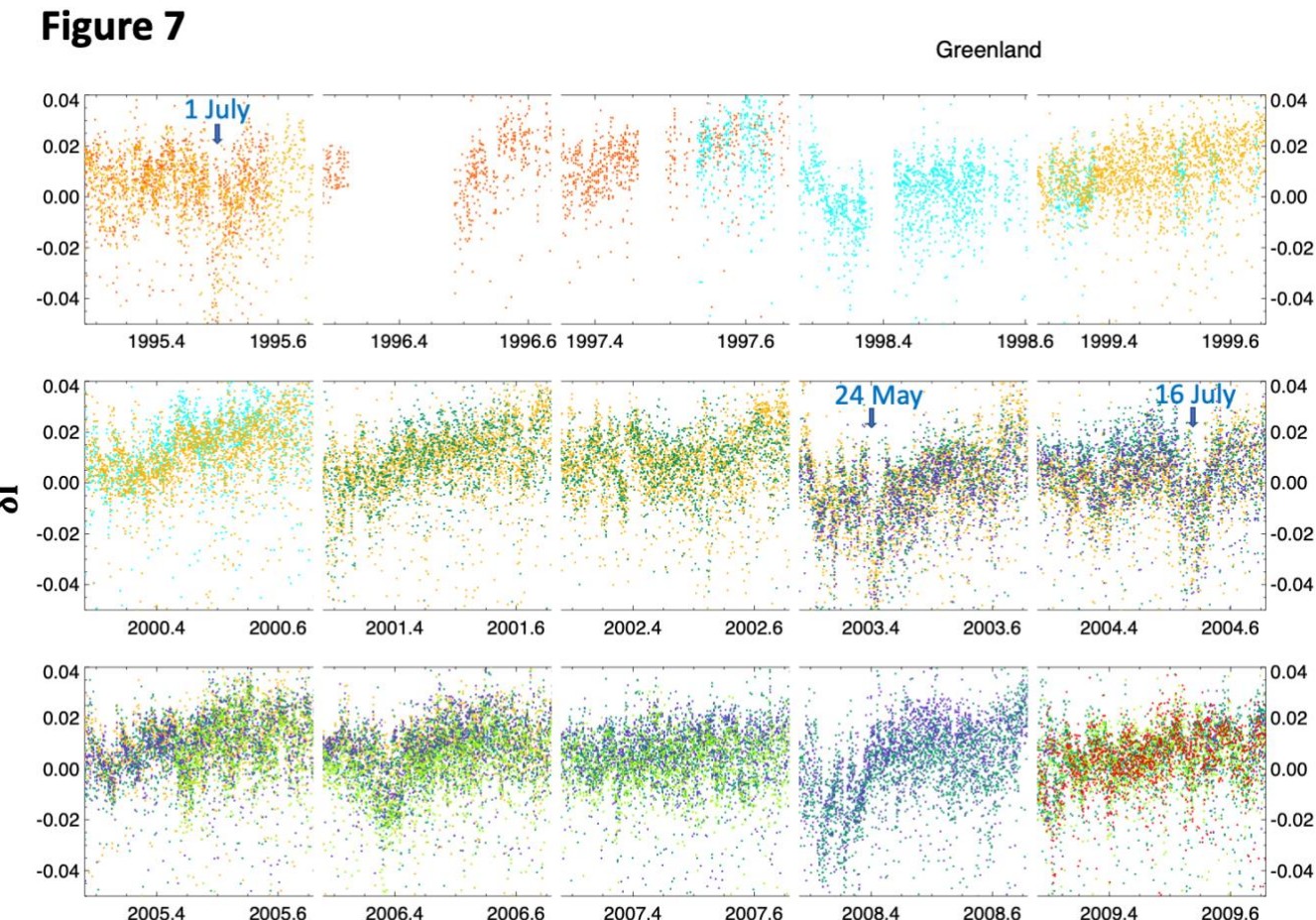

2 Figure 7. Time series of empirically calibrated δ**I** for all FOVs observed over Greenland for selected
3 years. Blue arrows indicate estimated dates when CO from boreal forest fires reach Greenland (see
4 text). Color scheme is same as other figures.