# Peer review of "Inter-Calibration of nine UV sensing instruments over Antarctica and Greenland since 1980"

_Atmospheric Measurement Techniques, 2020_

## Referee Comment (RC1) · Anonymous Referee #1 · 6 Apr 2020

General comments ———————— The paper "Inter-calibration of nine UV sensing instruments over Antarctica and Greenland since 1980" describes indeed what it announces while it also describes, and confirms, some episodic reductions caused by natural events in the last decades.

I recommend the paper for publication, the main reason being the importance of having a well-thought long-term satellite date record. That said, the paper is a bit too concise on some particular aspects, but these aspects can easily be improved/elaborated upon.

Specific comments ——————— a) On the fractional deviation delta_I. This variable occurs through the whole paper, but with different meanings: Of a particular measurement of a dark scene in Figure 1, and thereafter as some (summertime) average in Figure 2, but averaged per SZA bin in Figure 3. Different notations would be helpful

here. Then, the definition of Delta_I. It is in relation to a certain 4-term polynomial (is that 3rd order? If not, which polynomial orders?). Is it a constraint that the polynomial becomes zero at SAZ=90? In P5,L11 that is suggested, but is it enforced? I would expect a deviation with respect to the assumed 'truth' (see Figure 1) , so (I_obs - zeta(SZA))/zeta(SZA). That said, what is the reasoning behind the fractional/relative deviation (as opposed to absolute deviation)? Now measurements near zero reflection are weighted more heavily, and the expression may blow up (especially when having I_obs in the denominator, instead of zeta). Are low reflectance measurements more important? Note that the curve zeta itself, (P4L12) seems to be fitted by minimizing the absolute deviations (is that the case?) as standard for LS fitting.

Further on Figure 1, the cloud (especially of Greenland) seems to have more outliers below than above the polynomial. Why? Are the coefficients of the polynomial sensitive to these low outliers?

The delta_I is, as said, averaged over summertime. Does that mean that the 14/15 points of NOAA16 in Figure 2 are, on average, zero?

(NOAA-16 Seems the best choice for reference, but in P4L14 and P4L16, the lifitme is either 2001-2014 or 15 years. Both cannot be true.)

In Figure 6, the delta_I are averaged for each year, w.r.t. the satellites that were available for each year. That means that with only two satellites active (first year), the points are mirrored around zero. This graph which thus includes these mirroring properties in Figure 6 directly leads to the claim of the uncertainty of 0.35%. But this uncertainty should be different for each year, and years with many satellites should be weighted more than years with two satellites (like 1997) (?)

b) On adjusting the intensities. Section 4 starts with the claim that NOAA14 is low biased. How can that be seen in Figure 2? The light orange points do not lie below the NoaA16 points, nor do they lie below the y=0 horizontal line. Can you explain how we should interpret the graph, assuming that the claim is correct?

The strategy of inter-calibration works because at any time two or more instruments temporarily overlap (chaining). Is there some weighting of very early instruments in the process involved? Are there weak parts of the chain? Conversely, is the solution around 2007 (halfway NOAA16) better behaved than elsewhere?

Is it assumed or actively prescribed that the constant terms c0 are zero? It is assumed that all instruments were perfectly calibrated (no offsets). That might not be true. It does not automatically follow that, in this exercise, prescribing c0=0 would be necessary. Of course, it can be tried to allow for non-constant c0 in the inter-calibration. It would probably give better results to allow that freedom (lower residuals) , while necessitating some explaining (...)

Is it correct that the difference between Figure 4 w.r.t. Figure 2 is the correction of I with the gain factor in Table 1, following with the re-computation of delta_I?

On the remedy of the hysteresis (P9): So the first light observations of Nimbus-7 were removed. But the asociated observations of NOAA16 were not removed, so we do now compare (i.e. in the recomputing process to acquire Figure 4) different summertime averages of delta_I? Is that allowed?

c) On discussing the events.

The 1992 (P9L17) reduction: is it not visible for Greenland? Why not? (Aerosol transport?)

In P9L21, reductions are mentioned. When are they correlated (Greenland/Antarctica), and when not? And why? In general, the point you stress here is that the long-term drift is (just) insignificant, but the particular events are well observed by the satellites. That seems OK and well explained. On the other hand, you mention the Polashenski (2015) results to be also 0.05 per decade which is simular (P12L4). If it is insignificant, why mention it? (Can you explain the notation -0.05(0.06) in P11L20 ? )

d) On the graphs. More explaining of the graphs in the caption (in order to have more

self-explaining graphs) would be helpful (if that is allowed by the journal).

Technical corrections ——————————

P2L9: show |a| negligible long-term trend?

P5L19 stokes -> Stokes

P8L3: 'they' refers to?

P9L7: So the correction of Deland et al was not so good after all and by discarding these 9 minutes we got rid of all hysteresis by brute force (?) P9L17: multiple means 2 in this case.

P11L15 Figure 8 -> Figure 7.

P16: Might be an idea to extend the table with lifetime (start-end) per instrument. P16: c0 is - except for OMPS. Why? P16: consider setting c1 to 1 (without zeros) for nOAA16.

——————————————————————

---

## Referee Comment (RC2) · Anonymous Referee #2 · 13 Jul 2020

**General Comments:**

I recommend this paper for publication, pending consideration of the following comments. In general, the paper would benefit from just a few more details.

The figures should be improved significantly by increasing the font size.

**Specific Comments:**

**[Abstract, p. 2, lines 12–13]** *Three of these darkening events are explained by boreal*

[Figure]

*forest fires using trajectory modeling analysis.*
This is true, but this sentence implies that trajectory analysis is a part of this work, which it is not. Suggest removing "using trajectory analysis".

**[Sect. 1, p. 3, lines 5–6]** *This paper details the first step: the inter- calibration of radiances from the suite of nadir viewing instruments.*
What are the next steps? It would be nice if these were summarized, if even in a single sentence, to provide context.

**[Sect. 2, p. 4, lines 5–7]** *Rather than calibrate these additional instruments with a radiative transfer model using LER, we use an empirical approach to remove the solar zenith angle dependence on intensity.*
Although Sect. 3 goes on to explain the empirically based inter-calibration, this statement leaves me wondering why this was chosen over radiative transfer modeling. A simple statement here would establish a context for Sect. 3.

**[Sect. 3, p. 5, lines 12–14]** *One needs to pay particular attention to make sure the $\theta_0$ used is exactly simultaneous with the intensity, since the SBUV instruments have a different $\theta_0$ for each wavelength.*
A bit more explanation is needed here. These are scanning instruments, and as such wavelengths are not measured simultaneously or at precisely the same solar zenith angle. Is this not reflected in the data product files? Why is particular attention required?

**[Sect. 3, p. 5, line 19]** *. . . of 663 hPa.*
Why was 663 hPa chosen?
**[Sect. 3, p. 6, line 3]** *. . . but the $\delta I$ was still too dependent on $\theta_0$ . . .*
What is meant by this? Simply that the slope derived from VLIDORT (as in Figure 1) was too shallow?

**[Sect. 3, p. 6, lines 5–6]** *. . . Jaross et al (2008). They account for the snow BRDF which we omit.*
Would you quantify (at least to first order) what impact not accounting for snow BRDF would have on this analysis?

**[Sect. 4, p. 6, line 17]** *. . . NOAA-14 low biased compared to our reference (Figure 2).*
Maybe I am struggling with the color scheme in Figure 2, but NOAA-14 does not appear to be biased low to me. It appears to be positive at least half the time. Am I mis-reading the plot?

**[Sect. 4, p. 7, line 2]** *After adjustment, the biases are negligible (right panel Figure 3a).*
How is "negligible" defined in this context? Yes, the biases have been reduced, but are they now statistically insignificant? I suggest a different word be used or explained more precisely.

Also, this statement seems out of place. It should come *after* the adjustment is described in the following paragraphs. Here, you could lead with a statement about why adjustment is necessary.

**[Sect. 4, p. 7, line 4]** *To adjust intensities for a specific instrument a multiplicative factor ($c_1$) is chosen so that . . .*
Is the only reason that the additive coefficient, $c_0$, is not considered because of the

PMT zero-offset bias mentioned on p. 5? Is this adequately justified? If so, it would be worth stating here.

**[Sect. 4, p. 8, line 3]** *. . . they are not used in the intercalibration, but are used in the later trend analysis.*
Are "they" the data affected by the grating drive position errors (presumably corrected for in the trend analysis)? If so, please clarify. And why were they not used (I assume to remove any possibility of contamination)? Please clarify this as well.

**[Sect. 4, p. 8, lines 9–10]** *It is disconcerting that our correction does not bring them in closer alignment.*
If the authors themselves are disconcerted, then I certainly am. Would you please speculate as to why the correction does not improve agreement? What could this mean for the analysis?

**[Sect. 4, p. 10, lines 2–3]** *. . . this merged time series is the geophysical contribution.*
It might be more precise to say "this merged time series represents the geophysical contribution".

**[Sect. 5, p. 10, lines 17–18]** *For easier comparison we have transcribed the data from their Figure 4 onto our Figure 4c.*
It is still a somewhat difficult comparison in Figure 4. It would perhaps be clearer if the merged time series were compared to MODIS in a dedicated plot.

**[Sect. 5, p. 11, line 15]** *. . . on those dates (Figure 8).*
I don't see a Figure 8 in the manuscript. Or is this referring to Damoah et al. (2004)?

**[Sect. 6, p. 13, lines 7–9]** *These calibrated intensities will be used to derive a UV cloud albedo record over the tropics and midlatitudes since 1980.*
Again, how will these be used to derive a UV cloud albedo record?

**Typos:**

**[Abstract, p. 2, line 9]** *While the calibrated intensities show negligible long-term trend over Antarctica, . . .*
Add "a" before "negligible".

**[Sect. 1, p. 2, line 19]** *. . . deployed a suit of SBUV-2 instruments on board . . .*
"suit" should be "suite".

---

## Author Comment (AC2) · 17 Aug 2020

The figures should be improved significantly by increasing the font size. Font size has been increased

[Abstract, p. 2, lines 12–13] Three of these darkening events are explained by boreal forest fires using trajectory modeling analysis. This is true, but this sentence implies that trajectory analysis is a part of this work, which it is not. Suggest removing "using trajectory analysis". Modified abstract, thank you [Sect. 1, p. 3, lines 5–6] This paper details the first step: the inter- calibration of radiances from the suite of nadir viewing instruments. What are the next steps? It would be nice if these were summarized, if even in a single sentence, to provide context. Added text. "The second step retrieves

a Black-sky cloud albedo (BCA) record from the inter-calibrated intensities (Weaver et al. 2020) and compares the BCA with the Shortwave CERES cloud albedo." [Sect. 2, p. 4, lines 5–7] Rather than calibrate these additional instruments with a radiative transfer model using LER, we use an empirical approach to remove the solar zenith angle dependence on intensity. Although Sect. 3 goes on to explain the empirically based inter-calibration, this statement leaves me wondering why this was chosen over radiative transfer modeling. A simple statement here would establish a context for Sect. 3. I reworked the text. "At first glance the VLIDORT simulation appears to simulate the observations (red trace Figure 1) and we considered using the I to  relationship simulated by VLIDORT as a reference (instead of using NOAA-16). But closer examination shows that the slope of the VLIDORT is shallow compared with the observations. The resulting $\delta$I would still be slightly dependent on  which would complicate the analysis." [Sect. 3, p. 5, lines 12–14] One needs to pay particular attention to make sure the $\theta$0 used is exactly simultaneous with the intensity, since the SBUV instruments have a different $\theta$0 for each wavelength. A bit more explanation is needed here. These are scanning instruments, and as such wavelengths are not measured simultaneously or at precisely the same solar zenith angle. Is this not reflected in the data product files? Why is particular attention required? I removed the sentence. [Sect. 3, p. 5, line 19] ...of 663 hPa. Why was 663 hPa chosen? It is the mean surface pressure for Antarctica [Sect. 3, p. 6, line 3] ...but the $\delta$I was still too dependent on $\theta$0 ... What is meant by this? Simply that the slope derived from VLIDORT (as in Figure 1) was too shallow? Yes, I reworked the text, see above. [Sect. 3, p. 6, lines 5–6] ...Jaross et al (2008). They account for the snow BRDF which we omit. Would you quantify (at least to first order) what impact not accounting for snow BRDF would have on this analysis? Added Paragraph. "Another, more sophisticated approach to validate sun-normalized radiances over ice sheets is described in Jaross et al. (2008). They account for snow surface BRDF and off-nadir viewing angles. Nadir 330nm reflectances simulated using their snow BRDF model are 1% less than those assuming a Lambertian surface at  =70o; disparities are near zero at  =50o.

Our nadir observed $\delta$I was not sensitive to solar azimuth angle over Antarctica."

[Sect. 4, p. 6, line 17] . . . NOAA-14 low biased compared to our reference (Figure 2). Maybe I am struggling with the color scheme in Figure 2, but NOAA-14 does not appear to be biased low to me. It appears to be positive at least half the time. Am I mis-reading the plot? Oops, My error - corrected the text

[Sect. 4, p. 7, line 2] After adjustment, the biases are negligible (right panel Figure 3a). How is "negligible" defined in this context? Yes, the biases have been reduced, but are they now statistically insignificant? I suggest a different word be used or explained more precisely. See below. Also, this statement seems out of place. It should come after the adjustment is described in the following paragraphs. Here, you could lead with a statement about why adjustment is necessary. Yes, good suggestion, Reworked text "The positive bias for NOAA-17 and 18 is consistent at all bins and suggests that a simple adjustment of the intensities might reduce these biases." [Sect. 4, p. 7, line 4] To adjust intensities for a specific instrument a multiplicative factor (c1) is chosen so that . . . Is the only reason that the additive coefficient, c0, is not considered because of the PMT zero-offset bias mentioned on p. 5? Is this adequately justified? If so, it would be worth stating here. Initially, my results showed a non zero offset bias. Scientist/Engineers at NASA and SSAI (Science Systems and Applications) have been working for years to improve the calibration of the SBUV radiances to retrieve accurate ozone products; so they are very familiar with the instruments. They told me that radiances from the PMT can't have a non zero offset; rather what I was seeing are non-linearities at low signal levels. [Sect. 4, p. 8, line 3] . . . they are not used in the intercalibration, but are used in the later trend analysis. Are "they" the data affected by the grating drive position errors (presumably corrected for in the trend analysis)? If so, please clarify. And why were they not used (I assume to remove any possibility of contamination)? Please clarify this as well. Yes, the other review picked this up too; text has been clarified. Thank you

[Sect. 4, p. 8, lines 9–10] It is disconcerting that our correction does not bring them

in closer alignment."If the authors themselves are disconcerted, then I certainly am. Would you please speculate as to why the correction does not improve agreement? What could this mean for the analysis? Perhaps the estimated grating positions are wrong. I don't know what else to do. Further work needs to be done on this disparity. [Sect. 4, p. 10, lines 2–3] . . . this merged time series is the geophysical contribution. It might be more precise to say "this merged time series represents the geophysical contribution". Yes that's better Thank you [Sect. 5, p. 10, lines 17–18] For easier comparison we have transcribed the data from their Figure 4 onto our Figure 4c."It is still a somewhat difficult comparison in Figure 4. It would perhaps be clearer if the merged time series were compared to MODIS in a dedicated plot. [Sect. 5, p. 11, line 15] . . . on those dates (Figure 8)."I don't see a Figure 8 in the manuscript. Or is this referring to Damoah et al. (2004)? Should be Figure 7, text has been corrected. [Sect. 6, p. 13, lines 7–9] These calibrated intensities will be used to derive a UV cloud albedo record over the tropics and midlatitudes since 1980."Again, how will these be used to derive a UV cloud albedo record? Added addition sentence, see above Typos: [Abstract, p. 2, line 9] While the calibrated intensities show negligible long-term trend over Antarctica, . . ."Add "a" before "negligible". text has been corrected. [Sect. 1, p. 2, line 19] . . . deployed a suit of SBUV-2 instruments on board . . . "suit" should be "suite". text has been corrected.
* * *

---

## Author Response (AR1)

On the fractional deviation delta\_I. This variable occurs through the whole paper, but with different meanings: Of a particular measurement of a dark scene in Figure 1, and thereafter as some (summertime) average in Figure 2, but averaged per SZA bin in Figure 3. Different notations would be helpful.

The fractional deviation delta\_I that is used throughout the paper is always from Equation 1. The only difference is how it is averaged.

Then, the definition of Delta\_I. It is in relation to a certain 4-term polynomial (is that 3rd order? If not, which polynomial orders?).

I am actually using a 5-degree polynomial (6 term). I have corrected the text and figures. Thanks for catching this.

Is it a constraint that the polynomial becomes zero at SAZ=90? In P5,L11 that is suggested, but is it enforced?

There is no constraint that the polynomial is zero at SZA=90. Added text "Although the polynomial fit is not constrained to have **I**obs=0 at a solar zenith angle of 900, it appears so, consistent with this instrument design (Figure 1). "

I would expect a deviation with respect to the assumed 'truth' (see Figure 1), so (I\_obs - zeta(SZA))/zeta(SZA). I checked my IDL code and the fractional deviation is calculated as (I\_obs - zeta(SZA))/zeta(SZA), text and figures have been corrected. Thanks for catching this.

That said, what is the reasoning behind the fractional/relative deviation (as opposed to absolute deviation)? Now measurements near zero reflection are weighted more heavily, and the expression may blow up (especially when having I\_obs in the denominator, instead of zeta). We chose this definition because we are ultimately interested in the percentage error in the intensity.

Are low reflectance measurements more important? Note that the curve zeta itself, (P4L12) seems to be fitted by minimizing the absolute deviations (is that the case?) as standard for LS fitting. Yes, the fitted polynomial Figure 1 minimize the absolute deviations.

Further on Figure 1, the cloud (especially of Greenland) seems to have more outliers below than above the polynomial. Why?

The outliers below the polynomial fit (especially over Greenland) are from scenes that have absorbing material (dust, black carbon) in the satellite FOV. Most of the scenes, especially over Antarctica, are free of absorbing material and are at the upper limit of their reflectivity. The ice can get darker but can't get any brighter.

Are the coefficients of the polynomial sensitive to these low outliers? To maintain simplicity we included these outliers in the polynomial fit.

The delta\_I is, as said, averaged over summertime. Does that mean that the 14/15 points of NOAA16 in Figure 2 are, on average, zero? The average of all the NOAA-16 scenes will be zero but the annual averages shown in Figure 2 may not exactly be zero.

(NOAA-16 Seems the best choice for reference, but in P4L14 and P4L16, the lifitme is either 2001-2014 or 15 years. Both cannot be true.) Should read 14 year, text corrected.

In Figure 6, the delta\_I are averaged for each year, w.r.t. the satellites that were available for each year. That means that with only two satellites active (first year), the points are mirrored around zero. This graph which thus includes these mirroring properties in Figure 6 directly leads to the claim of the uncertainty of 0.35%. But this uncertainty should be different for each year, and years with many satellites should be weighted more than years with two satellites (like 1997) (?) At some point we were doing this. Calculating an uncertainty for each year based on the uncertainty of each individual instrument. In the interest of simplicity, we chose not to present this more complicated approach.

b) On adjusting the intensities. Section 4 starts with the claim that NOAA14 is low biased. How can that be seen in Figure 2? The light orange points do not lie below the NoaA16 points, nor do they lie below the y=0 horizontal line. Can you explain how we should interpret the graph, assuming that the claim is correct? Oops, My error. I corrected the text

The strategy of inter-calibration works because at any time two or more instruments temporarily overlap (chaining). Is there some weighting of very early instruments in the process involved? Are there weak parts of the chain? Conversely, is the solution around 2007 (halfway NOAA16) better behaved than elsewhere? There is no explicit weighting of earlier instruments in the slope or uncertainty determination. I would speculate that 1997 (at least over Antarctica) is weakest part chain because of grating drive errors.

Is it assumed or actively prescribed that the constant terms c0 are zero? It is assumed that all instruments were perfectly calibrated (no offsets). That might not be true. It does not automatically follow that, in this exercise, prescribing c0=0 would be neces- sary. Of course, it can be tried to allow for non-constant c0 in the inter-calibration. It would probably give better results to allow that freedom (lower residuals), while necessitating some explaining (...)

Initially, I allowed c0 to vary but was later told by those with intimate knowledge of the SBUV instruments that a non-zero offset not possible with the instruments photomultiplier tubes.

Is it correct that the difference between Figure 4 w.r.t. Figure 2 is the correction of I with the gain factor in Table 1, following with the re-computation of delta\_I? Yes On the remedy of the hysteresis (P9): So the first light observations of Nimbus-7 were removed. But the asociated observations of NOAA16 were not removed, so we do now compare (i.e. in the recomputing process to acquire Figure 4) different summertime averages of delta\_I? Is that allowed?

The last two panels of Figure 5 show the solar zenith angle delta\_I relationship for NOAA-16 (dark traces). Neither show increasing delta\_I with decreasing SZA that Nimbus-7 shows. This is a sign of hysteresis. In fact none of the other instruments show this feature, they are insensitive to SZA.

c) On discussing the events.

The 1992 (P9L17) reduction: is it not visible for Greenland? Why not? (Aerosol transport?) We don't know for sure but yes, probably differences in Sulfate aerosol transport.

In P9L21, reductions are mentioned. When are they correlated (Greenland/Antarctica), and when not? And why? The darkening events are from regional transport of light-absorbing particles to the FOV of the satellite so we don't expect a coordinated simultaneous response over Greenland and Antarctica

In general, the point you stress here is that the long-term drift is (just) insignificant, but the particular events are well observed by the satellites. That seems OK and well explained. On the other hand, you mention the Polashenski (2015) results to be also 0.05 per decade which is simular (P12L4). If it is insignificant, why mention it? (Can you explain the notation -0.05(0.06) in P11L20?)

We mention the Polashenski result because it is an entirely different kind of measurement (in situ). I have added +- before 0.06 to clarify that that it is uncertainty

d) On the graphs. More explaining of the graphs in the caption (in order to have more self-explaining graphs) would be helpful (if that is allowed by the journal).

Technical corrections —————

P2L9: show |a| negligible long-term trend? Done

P5L19 stokes -> Stokes Done

P8L3: 'they' refers to? Done clarified text

P9L7: So the correction of Deland et al was not so good after all and by discarding these 9 minutes we got rid of all hysteresis by brute force (?) No, that's not really fair to Matt. There are uncertainties associated with Deland's calibration. My changes are within these uncertainties. He is familiar with my results.

P9L17: multiple means 2 in this case. Three instruments: NOAA-9 -11 and -14 had grating drive errors.

P11L15 Figure 8 -> Figure 7. Thank you

P16: Might be an idea to extend the table with lifetime (start-end) per instrument. P16: c0 is - except for OMPS. Why? P16: consider setting c1 to 1 (without zeros) for nOAA16. Fixed OMPS and NOAA-16.

**Anonymous Referee #2**

The figures should be improved significantly by increasing the font size. Font size has been increased

[Abstract, p. 2, lines 12–13] *Three of these darkening events are explained by boreal forest fires using trajectory modeling analysis.* This is true, but this sentence implies that trajectory analysis is a part of this work, which it is not. Suggest removing "using trajectory analysis". Modified abstract, thank you

[Sect. 1, p. 3, lines 5–6] This paper details the first step: the inter- calibration of radiances from the suite of nadir viewing instruments. [F] What are the next steps? It would be nice if these were summarized, if even in a single sentence, to provide context. Added text. "The second step retrieves a Black-sky cloud albedo (BCA) record from the inter-calibrated intensities (Weaver et al. 2020) and compares the BCA with the Shortwave CERES cloud albedo."

[Sect. 2, p. 4, lines 5–7] Rather than calibrate these additional instruments with a radiative transfer model using LER, we use an empirical approach to remove the solar zenith angle dependence on intensity. Sep Although Sect. 3 goes on to explain the empirically based inter-calibration, this statement leaves me wondering why this was chosen over radiative transfer modeling. A simple statement here would establish a context for Sect. 3. I reworked the text. "At first glance the VLIDORT simulation appears to simulate the observations (red trace Figure 1) and we considered using the I to  $\theta_o$  relationship simulated by VLIDORT as a reference (instead of using NOAA-16). But closer examination shows that the slope of the VLIDORT is shallow compared with the observations. The resulting  $\delta I$  would still be slightly dependent on  $\theta_o$  which would complicate the analysis."

[Sect. 3, p. 5, lines 12–14] One needs to pay particular attention to make sure the  $\theta_0$  used is exactly simultaneous with the intensity, since the SBUV instruments have a different  $\theta_0$  for each wavelength. [1] A bit more explanation is needed here. These are scanning instruments, and as such wavelengths are not measured simultaneously or at precisely the same solar zenith angle. Is this not reflected in the data product files? Why is particular attention required? I removed the sentence.

[Sect. 3, p. 5, line 19] ... of 663 hPa. Why was 663 hPa chosen? It is the mean surface pressure for Antarctica

[Sect. 3, p. 6, line 3] ... but the  $\delta$ I was still too dependent on  $\theta_0$  ... EP What is meant by this? Simply that the slope derived from VLIDORT (as in Figure 1) was too shallow? Yes, I reworked the text, see above.

[Sect. 3, p. 6, lines 5–6] ... Jaross et al (2008). They account for the snow BRDF which

we omit. Would you quantify (at least to first order) what impact not accounting for snow BRDF would have on this analysis? Added Paragraph. "Another, more sophisticated approach to validate sun-normalized radiances over ice sheets is described in Jaross et al. (2008). They account for snow surface BRDF and off-nadir viewing angles. Nadir 330nm reflectances simulated using their snow BRDF model are 1% less than those assuming a Lambertian surface at  $\theta_0 = 70_0$ ; disparities are near zero at  $\theta_0$ = 500. Our nadir observed **\delta I** was not sensitive to solar azimuth angle over Antarctica."

[Sect. 4, p. 6, line 17] . . . *NOAA-14 low biased compared to our reference (Figure 2).* Maybe I am struggling with the color scheme in Figure 2, but NOAA-14 does not appear to be biased low to me. It appears to be positive at least half the time. Am I mis-reading the plot? Oops, My error - corrected the text

[Sect. 4, p. 7, line 2] After adjustment, the biases are negligible (right panel Figure 3a). [FP] How is "negligible" defined in this context? Yes, the biases have been reduced, but are they now statistically insignificant? I suggest a different word be used or explained more precisely. See below.

Also, this statement seems out of place. It should come *after* the adjustment is described in the following paragraphs. Here, you could lead with a statement about why adjustment is necessary. Yes, good suggestion, Reworked text "*The positive bias for NOAA-17 and 18 is consistent at all*  $\theta_o$  *bins and suggests that a simple adjustment of the intensities might reduce these biases.*"

[Sect. 4, p. 7, line 4] To adjust intensities for a specific instrument a multiplicative factor  $(c_1)$  is chosen so that . . . Second Second

[Sect. 4, p. 8, line 3] ... they are not used in the intercalibration, but are used in the later trend analysis. SepAre "they" the data affected by the grating drive position errors (presumably corrected for in the trend analysis)? If so, please clarify. And why were they not used (I assume to remove any possibility of contamination)? Please clarify this as well. Yes, the other review picked this up too; text has been clarified. Thank you

[Sect. 4, p. 8, lines 9–10] It is disconcerting that our correction does not bring them in *closer alignment*.[1] If the authors themselves are disconcerted, then I certainly am. Would you please speculate as to why the correction does not improve agreement? What could

this mean for the analysis? Perhaps the estimated grating positions are wrong. I don't know what else to do. Further work needs to be done on this disparity.

[Sect. 4, p. 10, lines 2–3] . . . this merged time series is the geophysical contribution. It might be more precise to say "this merged time series represents the geophysical contribution". Yes that's better Thank you

[Sect. 5, p. 10, lines 17–18] For easier comparison we have transcribed the data from their Figure 4 onto our Figure 4c. [FP] It is still a somewhat difficult comparison in Figure 4. It would perhaps be clearer if the merged time series were compared to MODIS in a dedicated plot.

[Sect. 5, p. 11, line 15] . . . on those dates (Figure 8). [F] I don't see a Figure 8 in the manuscript. Or is this referring to Damoah et al. (2004)? Should be Figure 7, text has been corrected.

[Sect. 6, p. 13, lines 7–9] These calibrated intensities will be used to derive a UV cloud albedo record over the tropics and midlatitudes since 1980.[1] Again, how will these be used to derive a UV cloud albedo record? Added addition sentence, see above

**Typos:**

[Sect. 1, p. 2, line 19] ... deployed a suit of SBUV-2 instruments on board ... "suit" should be "suite". text has been corrected.

| Inter-Calibration of nine UV sensing instruments over Antarctica and Greenland since 1980                                                      |
|------------------------------------------------------------------------------------------------------------------------------------------------|
| Clark Wester 12 D. K. Dhartiel, Dana I. West, Canden Labourt 4, David Heffern 14                                                               |
| Clark weaver 12 , P. K. Bhartia 2 , Dong L. wu 9 , Gordon Labow 3,1 , David Hallner 3,1 |
| 1 Atmospheric Chemistry and Dynamics Branch, NASA Goddard Space Flight Center, Greenbelt, MD                                        |
| 20771, USA.                                                                                                                                    |
| 2 Earth System Science Interdisciplinary Center (ESSIC), University of Maryland                                                     |
| College Park, MD 20742, USA                                                                                                                    |
| 3 Climate and Radiation Laboratory, NASA Goddard Space Flight Center, Greenbelt, MD 20771, USA.                                     |
| 4 Science Systems and Applications (SSAI), Inc., Lanham, MD 20706, USA.                                                             |
| Correspondence to: Clark Weaver clark i weaver@pasa gov                                                                                        |
|                                                                                                                                                |

Style Definition: Normal: Font: (Default) Times New Roman

**1 Abstract**

[revised manuscript text omitted]

| 1  | calculating a fractional deviation in terms of intensity ( $\delta I$ ) from Equation 1. For example, the right                                     |                                                                                                                                                                                                                       |
|----|-----------------------------------------------------------------------------------------------------------------------------------------------------|-----------------------------------------------------------------------------------------------------------------------------------------------------------------------------------------------------------------------|
| 2  | panel of Figure 1 shows an anomalously low intensity sampled over a dark scene $(I_{obs}^{dark \ scene})$                                           |                                                                                                                                                                                                                       |
| 3  | observed at a solar zenith angle ( $\theta_o^{\text{dark scene}}$ ); it is compared with the intensity that NOAA-16 would                           |                                                                                                                                                                                                                       |
| 4  | likely have observed at that solar zenith angle ( $\xi(\theta_0^{\text{dark scene}})$ ). The difference is divided by $\xi(\theta_0^{\text{dark}})$ | Deleted: Iobs                                                                                                                                                                                                         |
| 5  | scene ) to produce a fractional deviation in intensity $\delta I$ which is common throughout the manuscript.                                 |                                                                                                                                                                                                                       |
| 6  | $\delta \mathbf{I} = \frac{\mathbf{I}_{obs} - \xi(\theta_o)}{\xi(\theta_o)}$ Equation 1                                                             | Deleted: $\delta \mathbf{I} = \frac{\mathbf{I}_{obs} - \xi(\theta_o)}{\mathbf{I}_{obs}}$                                                                                                                              |
| 7  | Each UV instrument has its own unique $I_{obs}$ to $\theta_o$ relationship mainly because the photomultiplier tube                                  |                                                                                                                                                                                                                       |
| 8  | (PMT) for each instrument has a slightly different response function. The underlying scene UV albedo                                                |                                                                                                                                                                                                                       |
| 9  | (averaged over an instrument's lifetime) could be slightly different for each instrument, which would                                               |                                                                                                                                                                                                                       |
| 10 | also change the $I_{obs}$ to $\theta_o$ relationship, but we expect the Antarctic plateau albedo to be stable over time.                            |                                                                                                                                                                                                                       |
| 11 | The SBUV PMTs are designed to have a zero-offset bias, i.e. zero current response when there are zero                                               |                                                                                                                                                                                                                       |
| 12 | photon counts. Although the polynomial fit is not constrained to have $I_{obs}=0$ at a solar zenith angle of                          | Deleted: Over Antarctica                                                                                                                                                                                              |
| 13 | $90^{\circ}$ it appears so consistent with this instrument design (Figure 1)                                                                        | Deleted: appears                                                                                                                                                                                                      |
| 14 | y , it appears so, consistent with this instrument design (Figure 1).                                                                        | Deleted: One needs to pay particular attention to make sure the $\theta_0$ used is exactly simultaneous with the intensity, since the SBUV instruments have a different $\theta_0$ for each wavelength. |
| 15 | We also show estimates of Intensity calculated by the radiative transfer model VLIDORT (Vector                                                      |                                                                                                                                                                                                                       |
| 16 | LInearized Discrete Ordinate Radiative Transfer package, Spurr, 2006). Here we assume Lambertian                                                    |                                                                                                                                                                                                                       |
| 17 | surface albedo of .95, and Rayleigh atmosphere with surface pressure of 663 hPa. The number of half-                                                |                                                                                                                                                                                                                       |
| 18 | space quadrature streams is 40; the number of Stokes vector parameters is 3. At first glance the                                             | Deleted: stokes                                                                                                                                                                                                       |

| 1 | VLIDORT simulation appears to simulate the observations, (red trace Figure 1) and we considered using       |
|---|-------------------------------------------------------------------------------------------------------------|
| 2 | the I to $\theta_0$ relationship simulated by VLIDORT as a reference (instead of using NOAA-16). But closer |
| 3 | examination shows that the slope of the VLIDORT is shallow compared with the observations. The              |
| 4 | resulting $\delta I$ would still be slightly dependent on $\theta_0$ which would complicate the analysis.   |
|   |                                                                                                             |

- 5 Another, more sophisticated approach to validate sun-normalized radiances over ice sheets is described
- 6 in Jaross et al. (2008). They account for snow surface BRDF and off-nadir viewing angles. Nadir
- 7 330nm reflectances simulated using their snow BRDF model are 1% less than those assuming a
- 8 Lambertian surface at  $\theta_0 = 70^\circ$ ; disparities are near zero at  $\theta_0 = 50^\circ$ . Our nadir observed  $\delta I$  is not
- 9 sensitive to solar azimuth angle over Antarctica.
- 10 The suite of SBUV/2 instruments provides nadir observations with a 170x170km Field Of View (FOV).
- 11 But the OMPS Mapper instrument has a smaller nominal 50x50km FOV, except at the two most nadir
- 12 viewing positions. Here the FOV widths are 20 and 30 km (Seftor el al 2017). For consistency, we only
- 13 used the Mapper viewing positions that were within a nadir-centered hypothetical 170x170km SBUV
- 14 FOV and aggregated their intensities (area weighted) prior to calculating  $\delta I$ . For each instrument we
- 15 calculate the summertime annual mean and plot the timeseries for both ice sheets (Figure 2).

**16 4. Adjusting the intensities**

- 17 The pre-calibrated intensities SBUV2 instruments on board NOAA-17, -18 and -19 appear to be high
- 18 biased compared to our reference (Figure 2). As described below, a cost-optimization approach is used
- 19 to adjust the intensities and reduce these disparities. Figure 2 only shows the summertime average  $\delta I$ ,
- 20 but when calibrating instruments, it is instructive to examine the  $\delta I$  dependence on  $\theta_0$  for individual

| Deleted: , but the slope of the simulation is slightly different than the observations (right panel of |
|---------------------------------------------------------------------------------------------------------------|
| Deleted: ). We tried                                                                                          |
| Deleted: ) but the                                                                                            |

|                           | Deleted: was                                                                                                                                      |
|---------------------------|---------------------------------------------------------------------------------------------------------------------------------------------------|
| (                         | Deleted: too                                                                                                                                      |
| $\gamma / \gamma$         | Moved down [1]: (2008).                                                                                                                           |
| $\left  \right\rangle$    | Deleted: complicated                                                                                                                              |
|                           | Deleted: Our method is similar but not as sophisticated as
the sun-normalized radiance validation approach described
in Jaross et al |
| $\langle \rangle \rangle$ | Deleted: They account for the snow BRDF which we omit.                                                                                            |
| - \ }                     | Moved (insertion) [1]                                                                                                                             |

[revised manuscript text omitted]